# Assessing the Drivers of Sustained Agricultural Economic Development in China: Agricultural Productivity and Poverty Reduction Efficiency

**Jianlin Wang [1,2]** , **Junbo Tong [3] and Zhong Fang [3,*]**

1   School of Finance, Fujian Business University, Fuzhou 350506, China; wangjianlin2023@fjbu.edu.cn
2   Min Merchants Research Center, Fuzhou 350506, China
3   School of Economics, Fujian Normal University, Fuzhou 350007, China; qsx20210013@student.fjnu.edu.cn
*   Correspondence: fazhong02@fjnu.edu.cn

**Abstract:** Poverty eradication is a crucial element of SDG 1. Whether the financial resources invested by the Government provide a critical impetus for deeply impoverished rural areas needs to be studied by quantitative analysis. Therefore, this study presents a theoretical analytical framework for agricultural production–government poverty reduction. It divides the poverty reduction process into two stages, agricultural production and poverty reduction, from the perspective of sustainable agricultural development. The comprehensive measurement and spatio-temporal evolution analysis of China's agricultural production and poverty reduction efficiency are conducted using a novel dynamic two-stage DEA model, which incorporates non-expected factors. The study found that (1) China's agricultural production and poverty reduction efficiency exhibit overall poor performance, characterized by two poles of differentiation. (2) The agricultural production efficiency score is higher in the northern region than in the southern region, while the poverty reduction efficiency score is higher in the eastern region compared to the western region. (3) The coupling and coordination between China's production efficiency and poverty reduction efficiency are inadequate.

**Keywords:** agricultural production; government poverty reduction; two-stage DSBM model; spatio-temporal evolution

## 1. Introduction

In recent years, due to the impact and influence of multiple factors such as global climate change, rapid urbanization, informatization, and marketization, global poverty reduction in agricultural production has shown increasing vulnerability [1]. Global poverty declined from 2015 to 2018, with the global poverty rate falling from 10.1% in 2015 to 8.6% in 2018. However, affected by the 2019 Coronavirus disease, the global poverty rate rose from 8.3% in 2019 to 9.2% in 2020, and in 2020, approximately 8 million workers worldwide fell into poverty [2]. On the Chinese side, over the past 40 years, the number of poor people in China has been reduced by roughly 800 million, and the rate of poverty reduction has been significantly higher than the global average over the same period. In 2020, China successfully addressed the issue of absolute poverty (defined as subsistence poverty, where individual and household incomes are insufficient to meet basic survival needs), lifting 98.99 million rural poor out of poverty and elevating 832 poverty-stricken counties. Nevertheless, this achievement does not signify the completion of China's poverty reduction endeavors. Relative poverty (defined as a situation in which an individual or a family has the resources to meet its basic needs, but its standard of living is far below the average level of society) is expected to persist for a prolonged period, and certain regions still confront the risk and challenge of individuals reverting to poverty following their upliftment.

Agricultural subsidies are one of the effective policy tools currently used by governments to support agricultural development [3]. Developing countries typically demonstrate relatively low per capita gross domestic product (GDP), lower levels of labor productivity, and relatively limited efficiency in poverty reduction through agricultural production. For many developing countries, agricultural production and government expenditures are central areas of concern for government decision-making. Fiscal efficiency is thus a crucial topic in public economics [4–8]. According to statistics from China's Ministry of Finance, the central government's financial allocation for poverty alleviation (the allocation of specific funds intended for rural poverty alleviation within the transfer payments from the central government's finance to local governments, following the division of responsibilities between them) has increased from CNY 66.1 billion in 2016 to CNY 146.1 billion in 2020, indicating an average annual growth rate of 17.2%. This rise has indirectly encouraged provinces, municipalities, and autonomous regions to invest over CNY 800 billion of local financial resources into poverty alleviation efforts. China's achievements in poverty alleviation have garnered widespread attention from scholars worldwide. Evaluating the productivity of Chinese agriculture and the efficiency of poverty reduction, as well as summarizing the lessons learned from China's poverty alleviation efforts, holds significant theoretical and practical importance.

Therefore, this paper will establish an analytical framework of agricultural production–government poverty reduction from the perspective of government expenditure to provide a comprehensive measure of the efficiency of poverty reduction in China's agricultural production. Based on the above analysis, this paper takes China as an example to explore the complex relationship between the development efficiency of the agricultural production subsystem and the development efficiency of the poverty reduction subsystem in the process of sustainable agricultural development in developing countries and to answer the following questions: (1) How do you objectively assess the poverty reduction efficiency of Chinese agricultural production? (2) How do you evaluate the coordinated performance between China's agricultural production efficiency and poverty reduction efficiency? (3) How can we effectively improve the production and poverty reduction efficiency of Chinese agriculture and promote the sustainable development of Chinese agriculture?

Addressing the aforementioned three issues, this paper initially models the output of sustainable agricultural development and establishes a novel two-stage theoretical framework for analyzing agricultural production efficiency and poverty reduction efficiency. Subsequently, it conducts a comprehensive assessment of agricultural production efficiency and poverty reduction efficiency in rural China while also contributing to theoretical advancements in the field of poverty research. To objectively assess the production and poverty reduction efficiency in Chinese agriculture, this study introduces the DSBM (dynamic slack-based measure) model and the network DEA (data envelopment analysis) model alongside the traditional DEA model. It constructs a two-stage improved DSBM model to evaluate agricultural production efficiency, poverty reduction efficiency, and the overall efficiency of agricultural production and poverty reduction. This enhanced model allows for the visualization of efficiency disparities among regions and stages. Moreover, it facilitates the direct calculation of the potential for enhancing production and poverty reduction efficiencies based on input–output indicators. Finally, leveraging empirical analysis of agricultural production poverty reduction across 27 provinces in China from 2016 to 2020, this study offers valuable insights into the existing status and coordination level of agricultural production poverty reduction in China. Through matrix analysis, the study identifies key areas for improvement and provides pertinent policy recommendations. These recommendations serve as a scientific foundation for decision-making processes, offering valuable guidance for developing countries striving to achieve sustainable agricultural development.

## 2. Literature Review

Sustainable agricultural development refers to agricultural progress that enhances resource efficiency, bolsters resilience, and ensures social equity and responsibility within agriculture and food systems [9]. Absolute poverty denotes a state of severe deprivation of fundamental human needs, including access to food, safe drinking water, sanitation, healthcare, housing, education, and information [10]. Relative poverty, on the other hand, occurs when individuals are considered poor due to their standard of living falling below the prevailing societal norm [11].

Sustained economic growth is the most fundamental driver of poverty reduction, and improvements in agricultural productivity and reductions in rural poverty are also important drivers of sustainable agricultural development [12]. However, economic growth reduces poverty based on the premise that average incomes rise while distribution changes in a pro-poor direction. Conversely, if economic growth is accompanied by worsening income inequality, poverty may increase rather than decrease. Because of this obvious negative effect of income disparity on poverty reduction, relevant scholars have also conducted a lot of research on the effect of factors such as distribution structure, transfer payments, and fiscal allocation on poverty reduction [13–15].

Much of the established literature has utilized DEA modeling to study the problem of agricultural production efficiency: Farrel (1957) was the first to use a linear programming approach to measure the frontier of agricultural production efficiency in the United Kingdom, which is considered to be the origin of the basic idea of the DEA approach [16]. Research on measuring and evaluating the efficiency of agricultural production and poverty reduction has focused on the following aspects:

(1) Research on factors influencing the efficiency of agricultural production for poverty reduction has been conducted extensively. Some scholars suggest that factors such as sustained economic growth, government support, land resources, and the utilization of agricultural machinery can significantly impact the effectiveness of poverty reduction in agricultural production [17–22]. Additionally, recent studies have explored the role of information technology and infrastructure investment in enhancing the efficiency of poverty reduction in agriculture. For example, Zhang et al., 2023 [23], argued that infrastructure projects can sustainably alleviate multidimensional poverty in the local community's post-project completion. Similarly, Hartwig et al., 2023 [24], found that investments in transportation and information and communication technology (ICT) infrastructure can enhance households' ability to cope with shocks.

(2) Research on the application of DEA methods in the evaluation of agricultural production efficiency and poverty reduction efficiency. For the DEA evaluation methods of agricultural production efficiency and poverty reduction efficiency, many scholars provide different methods. Pan et al., 2021 [25], used a combination of the data envelopment analysis (DEA) model and the Malmquist index to analyze agricultural production efficiency in the Yangtze River Economic Belt. When studying the efficiency of agricultural production in the Yangtze River Economic Zone, the impact of environmental variables cannot be ignored. Lun et al., 2021 [26], included the carbon emissions generated in the process of agricultural production as unintended outputs in the efficiency evaluation model. Hobbs et al., 2020 [27], included the amount of unsatisfactory aid as an unintended output, making it an important factor in evaluating rural ecological performance. Yang et al., 2022 [28], studied the poverty reduction efficiency of China's agricultural production using a network EBM model with unintended outputs.

The above literature provides diverse perspectives and methods to evaluate agricultural poverty reduction efficiency, serving as a crucial reference for assessing agricultural production and poverty reduction effectiveness. But, there is still some room for research on poverty reduction efficiency or poverty reduction effects. Some existing research focuses on the poverty reduction effects of economic growth and agricultural economic growth, as well as government fiscal expenditures or poverty alleviation programs. However, these

studies may not comprehensively consider the impacts of agricultural production and fiscal expenditures on poverty reduction. In addition, in terms of the use of DEA methodology, the existing literature mainly studies agricultural production efficiency or directly evaluates the efficiency of poverty reduction, and with the expansion of the DEA methodology, network DEA has begun to be commonly used in the field of efficiency evaluation.

Therefore, this paper proposes a two-stage dynamic spatio-temporal benefit model (DSBM) to comprehensively evaluate the efficiency of poverty reduction in Chinese agricultural production. The main contributions of this paper are as follows: (1) The inclusion of central financial poverty alleviation funds in the theoretical framework of China's agricultural production poverty reduction efficiency for the first time. The construction of the two-stage DSBM model offers a new research perspective for studying agricultural production poverty reduction efficiency. (2) Analyses of the dynamic spatio-temporal evolution law of China's agricultural production poverty reduction efficiency across regions and time. This measures the poverty reduction efficiency of China's agricultural production at both two-stage integrated system and single-stage subsystem levels. (3) Calculation of the efficiency optimization space of input–output indicators of production efficiency and poverty reduction efficiency.

This research fills a gap in the existing literature on assessing the process of China's sustainable agricultural development from agricultural production to poverty reduction, aiming to achieve evaluation results with higher accuracy. This contribution holds great significance for deepening research on the evaluation of agricultural production efficiency for poverty reduction and effectively enhancing China's agricultural production efficiency for poverty reduction.

## 3. Research Methodology

### 3.1. The Entropy Method

#### 3.1.1. Detailed Indicators

In the second stage of the model, the output items of the government's poverty reduction stage comprise five secondary indicators involving 22 specific sub-indicators (Figure 1). However, directly incorporating these indicators into the DEA model may pose unsolvable problems. Therefore, this model initially adopts Shannon's entropy method (1948) to calculate the value of each sub-indicator.

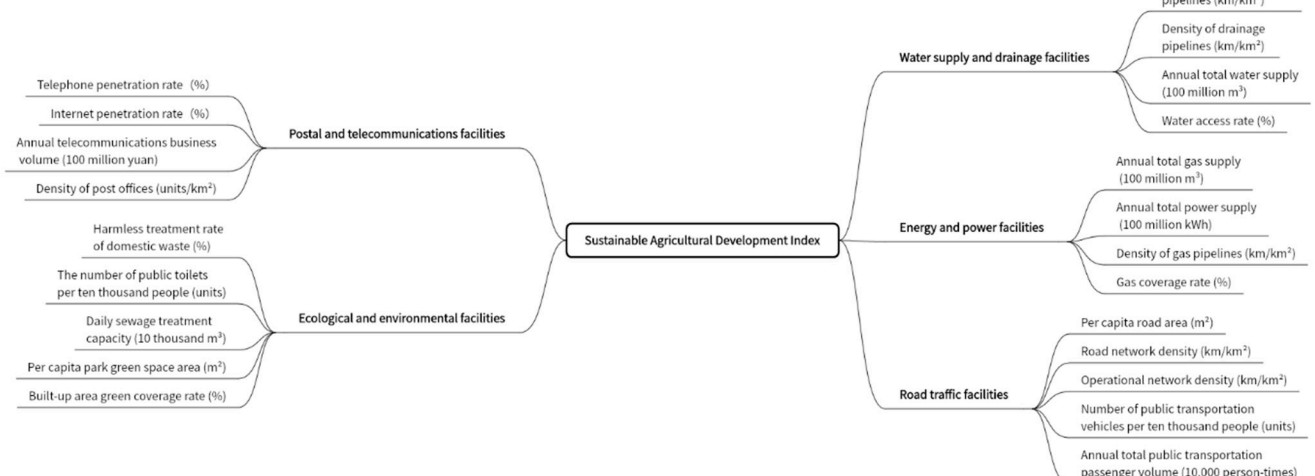

**Figure 1.** Sustainable Agricultural Development Index.

### 3.1.2. Entropy Method Steps

In the first step, to address the variability of different indicators, each indicator underwent standardization using the extreme value method. Positive and negative indicators were treated differently based on the nature of each indicator, following the Formula (1) below:

$$z_{ij} = \frac{x_{ij} - minX_j}{maxX_j - minX_j}, \quad z_{ij} = \frac{maxX_j - x_{ij}}{maxX_j - minX_j} \quad i = 1, 2, \cdots, n; \ j = 1, 2, \cdots, m. \quad (1)$$

In the second step, the entropy value $E_j$ is calculated as follows:

$$E_j = -ln \frac{1}{27} \sum_{i=1}^{27} \left[ \left( Z_{ij} / \sum_{i=1}^{n} Z_{ij} \right) ln \left( Z_{ij} / \sum_{i=1}^{n} Z_{ij} \right) \right]. \quad (2)$$

In the third step, the calculation of the coefficient of variation $D_j$ is as follows:

$$D_j = 1 - E_j. \quad (3)$$

In the fourth step, the weight of the $j$th index is calculated:

$$W_j = \frac{D_j}{\sum\limits_{j=1}^{n} D_j}. \quad (4)$$

The fifth step is obtaining the comprehensive score of the Sustainable Agriculture Development Index (SADI). After applying the entropy weight method to calculate the comprehensive weight of each indicator, the comprehensive score for each year is calculated using Formula (5).

$$G_j = \sum \left( W_i Y_{ij} \right). \quad (5)$$

$G_j$ in Equation (5) represents the composite sustainable agriculture development score for year $j$. $W_i$ denotes the composite weight of the $i$th indicator; $Y_{ij}$ denotes the value of the $i$th indicator in year $j$; $i = 1, 2, \cdots, n; j = 1, 2, \cdots, m$.

### 3.2. Network Dynamic SBM–DEA Model

Suppose there are n provinces in China, denoted by $DMU_j(j = 1, 2, \ldots, n)$. Each $DMU_j$ can be categorized into two sub-phases (i.e., agricultural production and agricultural poverty reduction stage) in a given period $t(t = 1, 2, \ldots, T)$. For the $o$th province, it produces $d$ outputs $v_{ho}^t(h = 1, 2, \ldots, d)$ in the agricultural production stage using $k$ capitals $c_{uo}^{(t-1,t)}(u = 1, 2, \ldots, k)$ carried over to period t in period $t - 1$, along with $m$ inputs $x_{io}^t(i = 1, 2, \ldots, m)$, to produce $d$ outputs $v_{ho}^t(h = 1, 2, \ldots, d)$ and $l$ intermediate outputs $z_{fo}^t(f = 1, 2, \ldots, l)$; its additional $p$ inputs $g_{eo}^t(e = 1, 2, \ldots, p)$ at the stage of poverty reduction in agriculture, producing $q$ desired outputs $y_{ro}^t(r = 1, 2, \ldots, q)$ and $w$ undesired outputs $u_{bo}^t(b = 1, 2, \ldots, w)$. The value of the total efficiency of agricultural production for poverty reduction in China is then determined as follows:

$$\theta_o^* = min \frac{\sum_{t=1}^{T} w^t \left[ \beta_1 \left( 1 - \frac{1}{m+k} \left( \sum_{i=1}^{m} \frac{s_{io}^{t-}}{x_{io}^t} + \sum_{i=1}^{k} \frac{s_{uo}^{(t-1,t)-}}{c_{uo}^{(t-1,t)}} \right) \right) + \beta_2 \left( 1 - \frac{1}{p+w} \left( \sum_{e=1}^{p} \frac{s_{eo}^{t-}}{g_{eo}^t} + \sum_{b=1}^{w} \frac{s_{bo}^{t-}}{u_{bo}^t} \right) \right) \right]}{\sum_{t=1}^{T} w^t \left[ \beta_1 \left( 1 + \frac{1}{d+l} \left( \sum_{h=1}^{d} \frac{s_{ho}^{t+}}{v_{ho}^t} + \sum_{f=1}^{l} \frac{s_{fo}^{t+}}{z_{fo}^t} \right) \right) + \beta_2 \left( 1 + \frac{1}{q} \sum_{r=1}^{q} \frac{s_{ro}^{t+}}{y_{ro}^t} \right) \right]}, \quad (6)$$

s.t.

$$x_{io}^t = \sum_{j=1}^n \gamma_j^t x_{ij}^t + s_{io}^{t-}, \forall i, \forall t,$$

$$c_{uo}^{(t-1,t)} = \sum_{j=1}^n \gamma_j^t c_{uj}^{(t-1,t)} + s_{uo}^{(t-1,t)-}, \forall u, t = 2, \ldots, T,$$

$$\sum_{j=1}^n \gamma_j^{t-1} c_{uj}^{(t-1,t)} = \sum_{j=1}^n \gamma_j^t c_{uj}^{(t-1,t)}, \forall u; t = 2, \ldots, T,$$

$$v_{ho}^t = \sum_{j=1}^n \gamma_j^t v_{hj}^t - s_{ho}^{t+}, \forall h, \forall t,$$

$$z_{fo}^t = \sum_{j=1}^n \gamma_j^t z_{fj}^t - s_{fo}^{t+}, \forall f, \forall t,$$

$$\sum_{j=1}^n \gamma_j^t z_{fj}^t = \sum_{j=1}^n \lambda_j^t z_{fj}^t, \forall f, \forall t,$$

$$u_{bo}^t = \sum_{j=1}^n \lambda_j^t p_{bj}^t + s_{bo}^{t-}, \forall b, \forall t,$$

$$g_{eo}^t = \sum_{j=1}^n \lambda_j^t g_{ej}^t + s_{eo}^{t-}, \forall e, \forall t,$$

$$y_{ro}^t = \sum_{j=1}^n \lambda_j^t y_{rj}^t - s_{ro}^{t+}, \forall r, \forall t,$$

$$\gamma_j^t, \lambda_j^t \geq 0, \forall j, \forall t,$$

$$s_{io}^{t-}, s_{uo}^{(t-1,t)-}, s_{ho}^{t+}, s_{fo}^{t+}, s_{bo}^{t-}, s_{eo}^{t-}, s_{ro}^{t+} \geq 0, \forall i, u, h, f, b, e, r,$$

where $s_{io}^{t-}, s_{uo}^{(t-1,t)-}, s_{bo}^{t-}$, and $s_{eo}^{t-}$ represent slack variables of input factors, $s_{ho}^{t+}, s_{fo}^{t+}$, and $s_{ro}^{t+}$ represent slack variables of output factors, $w^t \geq 0 (\forall t)$ represents the weight of time period $t$, and $\beta_1 \geq 0$ and $\beta_2 \geq 0$ represent the weight of the production stage and poverty reduction stage, respectively. It should be noted that in the objective function, the excess of bad output is calculated in the same way as the excess of input because they have the same characteristics as input; that is, the smaller, the more favorable [29].

Where $w^t$ denotes the weight of period $t$, $\beta_i$ denotes the weight of stage $i$, and, at the same time, satisfy $\sum_{t=1}^T w^t = 1$ and $\beta_1 + \beta_2 = 1$. Currently, there is no consensus in the academic community on determining the weights of each period and each sub-stage, but it is generally believed that the $t$ period has a greater impact on the efficiency of poverty reduction in provincial agricultural production than the $t-1$ period. Given that Tone and Tsutsui (2014) [30] put the $t$ period in first place and then reduced the weights of the $t-1, t-2, \ldots, t-n$ periods in order, this paper assumes that the relative importance of the period efficiency of poverty reduction in provincial agricultural production increases year by year from 2016 to 2020, and the period efficiency in 2020 makes the greatest contribution to the evaluation of the overall efficiency of poverty reduction in provincial agricultural production. Therefore, this paper assigns weights of $w^1 = 0.1$, $w^2 = 0.15$, $w^3 = 0.2$, $w^4 = 0.25$, and $w^5 = 0.3$ for 2016, 2017, 2018, 2019, and 2020, respectively, similar to Tone and Tsutsui (2014) and Zha et al. (2016) [30,31]. In terms of sub-stage weights, this paper assumes that the production and poverty reduction stages contribute equally to the efficiency of poverty reduction in provincial agricultural production; that is, the same weight is assigned to both sub-stages, i.e., $\beta_1 = \beta_2 = 0.5$.

In order to comprehensively evaluate the efficiency of poverty reduction and agricultural production in multiple periods, this paper calculates the overall efficiency of sub-stages (agricultural production stage and poverty reduction stage in multiple periods), the overall efficiency of a single period (i.e., the efficiency of agricultural production for poverty reduction in $t$ period), and the single-period sub-stage efficiency (the efficiency of sub-stages in $t$ period). The overall efficiency of the agricultural production stage and poverty reduction stage can be, respectively, defined as $\theta_{o1}^*$ and $\theta_{o2}^*$:

$$\theta_{o1}^* = \frac{\sum_{t=1}^T w^t \left(1 - \frac{1}{m+k} \left(\sum_{i=1}^m \frac{s_{io}^{t-}}{x_{io}^t} + \sum_{i=1}^k \frac{s_{uo}^{(t-1,t)-}}{c_{uo}^{(t-1,t)}}\right)\right)}{\sum_{t=1}^T w^t \left(1 + \frac{1}{d+l} \left(\sum_{h=1}^d \frac{s_{ho}^{t+}}{v_{ho}^t} + \sum_{f=1}^l \frac{s_{fo}^{t+}}{z_{fo}^t}\right)\right)}, \forall t, \tag{7}$$

$$\theta_{o2}^* = \frac{\sum_{t=1}^T w^t \left(1 - \frac{1}{p+w} \left(\sum_{e=1}^p \frac{s_{eo}^{t-}}{g_{eo}^t} + \sum_{b=1}^w \frac{s_{bo}^{t-}}{u_{bo}^t}\right)\right)}{\sum_{t=1}^T w^t \left(1 + \frac{1}{q} \sum_{r=1}^q \frac{s_{ro}^{t+}}{y_{ro}^t}\right)}, \forall t. \tag{8}$$

The efficiency of poverty reduction in agricultural production in the $t$ period is defined as $\theta_o^{t*}$:

$$\theta_o^{t*} = \frac{\beta_1\left(1 - \frac{1}{m+k}\left(\sum_{i=1}^m \frac{s_{io}^{t-}}{x_{io}^t} + \sum_{i=1}^k \frac{s_{uo}^{(t-1,t)-}}{c_{uo}^{(t-1,t)}}\right)\right) + \beta_2\left(1 - \frac{1}{p+w}\left(\sum_{e=1}^p \frac{s_{eo}^{t-}}{g_{eo}^t} + \sum_{b=1}^w \frac{s_{bo}^{t-}}{u_{bo}^t}\right)\right)}{\beta_1\left(1 + \frac{1}{d+l}\left(\sum_{h=1}^d \frac{s_{ho}^{t+}}{v_{ho}^t} + \sum_{f=1}^l \frac{s_{fo}^{t+}}{z_{fo}^t}\right)\right) + \beta_2\left(1 + \frac{1}{q}\sum_{r=1}^q \frac{s_{ro}^{t+}}{y_{ro}^t}\right)}, \forall t. \tag{9}$$

The efficiency of the agricultural production stage in the $t$ period is defined as $\theta_{o1}^{t*}$:

$$\theta_{o1}^{t*} = \frac{1 - \frac{1}{m+k}\left(\sum_{i=1}^m \frac{s_{io}^{t-}}{x_{io}^t} + \sum_{i=1}^k \frac{s_{uo}^{(t-1,t)-}}{c_{uo}^{(t-1,t)}}\right)}{1 + \frac{1}{d+l}\left(\sum_{h=1}^d \frac{s_{ho}^{t+}}{v_{ho}^t} + \sum_{f=1}^l \frac{s_{fo}^{t+}}{z_{fo}^t}\right)}, \forall t. \tag{10}$$

The efficiency of the poverty reduction stage in the $t$ period is defined as $\theta_{o2}^{t*}$:

$$\theta_{o2}^{t*} = \frac{1 - \frac{1}{p+w}\left(\sum_{e=1}^p \frac{s_{eo}^{t-}}{g_{eo}^t} + \sum_{b=1}^w \frac{s_{bo}^{t-}}{u_{bo}^t}\right)}{1 + \frac{1}{q}\sum_{r=1}^q \frac{s_{ro}^{t+}}{y_{ro}^t}}, \forall t. \tag{11}$$

## 4. Results and Discussion

### 4.1. Data and Variables

#### 4.1.1. Variable Explanation

As shown in Figure 2, this study adopted a perspective of sustainable agricultural development to address poverty reduction, dividing the process into two stages: the agricultural production stage and the government poverty reduction stage, then introducing a two-stage DSBM model. The data on the total sown area of crops, legal units in agriculture, fertilizer use, the number of large and medium-sized tractors in agriculture, and the gross agricultural product were obtained from the 2016–2020 China Rural Statistical Yearbook. Additionally, basic data on the local government's financial expenditures, the central government's financial poverty alleviation funds, the number of people with minimum subsistence guarantees for rural residents, and the level of infrastructure of the agricultural economy were sourced from the 2016–2020 China Statistical Yearbook.

Specific variables are described in detail below:

(1)  The total sown area of crops refers to the area actually sown or transplanted with crops. It primarily encompasses nine major categories: grain, cotton, oilseeds, sugar, hemp, tobacco, vegetables and melons, medicinal herbs, and other crops.

(2)  Agricultural legal entity refers to the number of agricultural and industrial activity units established in accordance with the law, formally registered, with their own name, organization, and premises. These entities are capable of independently assuming civil liability, maintaining independent accounting records, and engaging primarily in crop cultivation, forestry, animal husbandry, fishery, or agriculture-related services.

(3)  Fertilizer use refers to the quantity of fertilizer actually used in agricultural production during the year, including nitrogen fertilizer, phosphate fertilizer, potash fertilizer, and compound fertilizer.

(4)  Gross agricultural product refers to the total monetary value of all products derived from agriculture, forestry, animal husbandry, and fishery activities within a specified period, typically one year. It serves as an indicator of the overall scale and outcomes of agricultural production.

(5)  Per capita disposable income of rural residents refers to the total amount of final consumption expenditures and savings available to rural resident survey households. It represents the income that survey households can use for discretionary purposes. Disposable income encompasses cash income as well as income in kind.

(6) Fiscal expenditure for poverty reduction in agricultural production refers to the funds allocated in fiscal expenditure specifically directed towards supporting agricultural production or more closely associated with poverty reduction in agriculture. In this paper, it is computed based on the existing division of responsibilities between the central government and local governments, considering the financial funds allocated by the central government for poverty alleviation and the funds allocated by local governments for agricultural support.

(7) The number of rural residents covered by the minimum subsistence guarantee refers to the count of rural residents benefiting from the minimum subsistence guarantee system. This system, implemented by the Chinese Government, provides support to rural residents whose annual per capita net household income falls below the locally defined minimum subsistence guaranteed standard.

(8) The Sustainable Agriculture Development Index (SADI) is computed using the entropy method and comprises 5 secondary infrastructure indicators and 22 tertiary indicators.

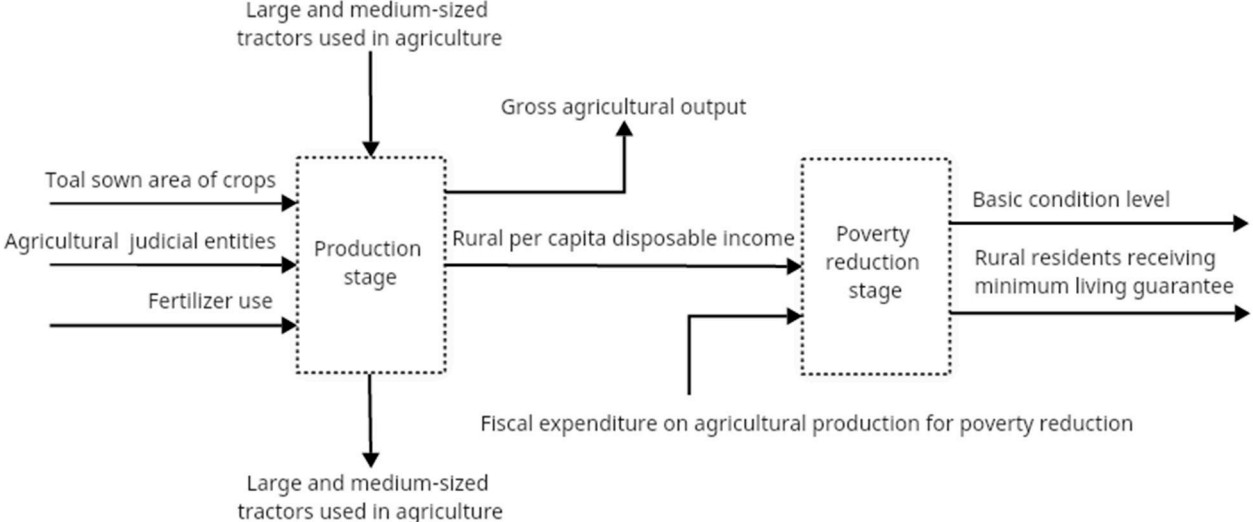

**Figure 2.** Theoretical analysis framework.

4.1.2. Data Description

As observed in Table 1, due to Beijing, Tianjin, and Shanghai not being covered by central government poverty reduction funds and significant data gaps in Tibet, this study focused on 27 provinces in Mainland China as the research subjects for analyzing the efficiency of poverty reduction in provincial agricultural production from 2016 to 2020.

For ease of discussing regional heterogeneity, we categorized the 27 provinces into three major regions: west, east, and central, as illustrated in Table 2.

*4.2. Empirical Results Analysis*

4.2.1. Network DSBM Model Performance Optimization Measure

This paper compared the single-stage SBM model and the two-stage network DSBM model to unveil the underestimated "black box" efficiency. As depicted in Table 3, the results demonstrate that the average total efficiency computed by the two-stage network DSBM model stands at 0.473, significantly higher than the overall efficiency of 0.239 observed in the single-stage SBM model, resulting in an efficiency release of 97.9%. Similar to the findings of the two-stage dynamic network model, only Hainan Province maintained its single-stage efficiency at the optimal frontier, while the single-stage efficiencies of the remaining 26 provinces declined. The two-stage network DSBM model provides a more realistic depiction of the agricultural production and poverty reduction process in China by unveiling the "black box" of traditional DEA.

**Table 1.** Input–output indicators of poverty reduction efficiency in China's agricultural production.

| Stage | Variable | Unit | Mean | Std. Dev. | Max | Min |
|---|---|---|---|---|---|---|
| **Agricultural Production stage** | | | | | | |
| Input | Total sown area of crops | thousand hectares | 6127.21 | 3717.95 | 14,910.13 | 553.54 |
| | Agricultural judicial entity | 10 thousand unit | 3.16 | 1.78 | 8.57 | 0.51 |
| | Fertilizer use | 10 thousand tons | 81.93 | 67.65 | 337.34 | 2.41 |
| Expected output | Gross agricultural output | CNY 100 million | 2294.99 | 1290.96 | 6244.84 | 155.52 |
| Link | Rural per capita disposable income | CNY 10 thousand | 1.42 | 0.40 | 3.19 | 0.75 |
| Carry-over | Large and medium-sized tractors used in agriculture | 10 thousand unit | 25.44 | 27.80 | 119.99 | 0.48 |
| **Poverty reduction stage** | | | | | | |
| Input | Local financial expenditure | CNY 100 million | 5862.20 | 3110.73 | 16,842.00 | 1174.81 |
| | Central government poverty reduction fund | CNY 100 million | 35.00 | 32.66 | 154.08 | 1.23 |
| Expected output | The level of agricultural economic infrastructure | / | 36.54 | 12.71 | 73.30 | 16.99 |
| Unexpected output | Rural residents receiving minimum living guarantee | 10 thousand people | 141.18 | 92.59 | 422.90 | 14.70 |

Note: As of 19 February 2024, CNY 100 ≈ USD 13.89.

**Table 2.** Distribution of the three major regions in China.

| Region | Provinces |
|---|---|
| West | Chongqing, Gansu, Ningxia, Qinghai, Guizhou, Yunnan, Shaanxi, Sichuan, Xinjiang |
| East | Guangxi, Hebei, Liaoning, Shandong, Jiangsu, Guangdong, Hainan, Fujian, Zhejiang |
| Central | Inner Mongolia, Henan, Anhui, Hubei, Hunan, Jilin, Jiangxi, Shanxi, Heilongjiang |

4.2.2. Analysis of Temporal Evolution Patter

Figure 3 depicts the overall efficiency values. As the figure shows, the performance of the comprehensive efficiency of China's agricultural production poverty reduction system was relatively stable during the observation period, showing a fluctuating upward trend, and then declined more sharply from 2016 to 2017, which was mainly affected by fluctuations in the efficiency of the agricultural poverty reduction subsystems. The efficiency of the agricultural poverty reduction system declined from 0.486 to 0.444 in 2016–2017, which is a decrease of 8.65%. Subsequently, China's agricultural production poverty reduction efficiency continued to rise in 2018–2020, characterized by a simultaneous increase in the efficiency of the agricultural production system and the agricultural poverty reduction system and the gradual emergence of coordination benefits. In particular, in 2020, the concluding year of China's poverty eradication battle, the comprehensive efficiency of agricultural production for poverty reduction increased significantly despite the huge impact of the epidemic, indicating that the Chinese Government's macro-control still has a strong influence on agricultural production and poverty reduction process.

**Table 3.** Average total efficiency of China's provinces from 2015 to 2019.

| Province | Production Efficiency | Poverty Reduction Efficiency | Overall Efficiency | Single-Stage Efficiency |
|---|---|---|---|---|
| Fujian | 0.230 | 1.000 | 0.615 | 0.512 |
| Guangdong | 0.285 | 0.687 | 0.486 | 0.469 |
| Zhejiang | 0.381 | 1.000 | 0.691 | 0.656 |
| Hainan | 1.000 | 1.000 | 1.000 | 1.000 |
| Jiangsu | 0.434 | 1.000 | 0.717 | 0.361 |
| Hebei | 0.316 | 0.295 | 0.305 | 0.088 |
| Liaoning | 0.644 | 0.442 | 0.543 | 0.184 |
| Shandong | 0.437 | 0.429 | 0.433 | 0.089 |
| Guangxi | 0.153 | 0.172 | 0.163 | 0.059 |
| Hubei | 0.306 | 0.246 | 0.276 | 0.105 |
| Heilongjiang | 1.000 | 0.440 | 0.720 | 0.112 |
| Hunan | 0.286 | 0.226 | 0.256 | 0.114 |
| Jilin | 1.000 | 0.405 | 0.703 | 0.142 |
| Jiangxi | 0.199 | 0.319 | 0.259 | 0.126 |
| Henan | 0.248 | 0.194 | 0.221 | 0.051 |
| Inner Mongolia | 1.000 | 0.405 | 0.702 | 0.117 |
| Anhui | 0.299 | 0.262 | 0.280 | 0.077 |
| Shanxi | 0.294 | 0.349 | 0.321 | 0.098 |
| Qinghai | 1.000 | 0.928 | 0.964 | 0.882 |
| Xinjiang | 1.000 | 0.432 | 0.716 | 0.215 |
| Sichuan | 0.217 | 0.195 | 0.206 | 0.099 |
| Chongqing | 0.105 | 0.390 | 0.247 | 0.121 |
| Guizhou | 0.140 | 0.164 | 0.152 | 0.067 |
| Ningxia | 0.794 | 0.929 | 0.862 | 0.398 |
| Yunnan | 0.418 | 0.171 | 0.295 | 0.075 |
| Shaanxi | 0.264 | 0.327 | 0.295 | 0.141 |
| Gansu | 0.505 | 0.182 | 0.343 | 0.091 |
| Mean | 0.480 | 0.466 | 0.473 | 0.239 |

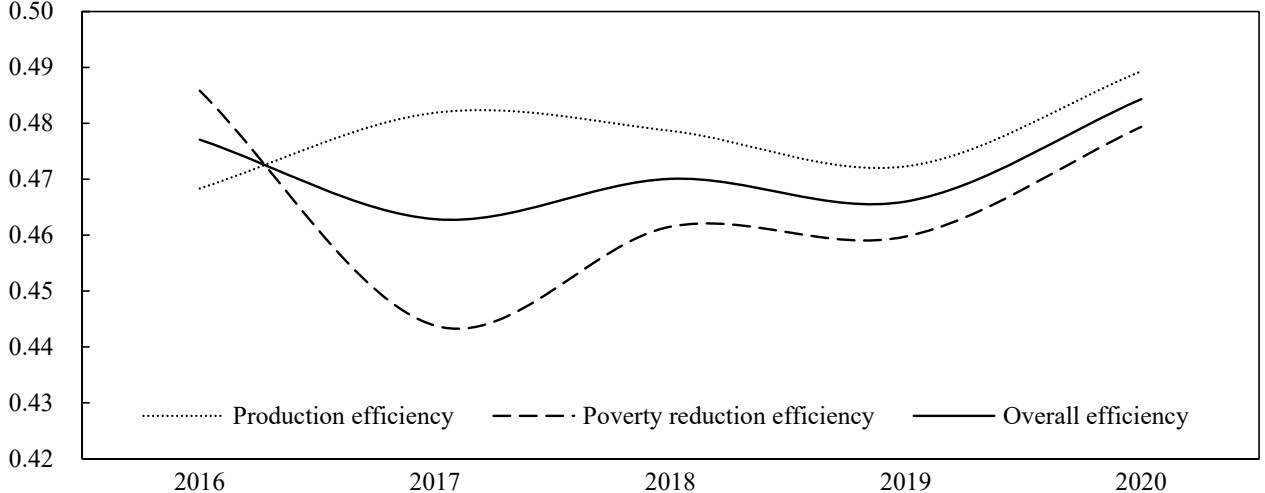

**Figure 3.** Trend of overall efficiency (2016–2020).

### 4.2.3. Analysis of Spatial Evolution Patterns

As Figure 4 shows, we categorized China's provincial agricultural production poverty reduction into two groups using the mean value of China's total efficiency of agricultural production poverty reduction as the reference point. Subsequently, we mapped the spatial distribution of China's agricultural production poverty reduction efficiency from 2016 to 2020. Regarding the growth trend, the agricultural production efficiency of most

provinces mainly shows a downward trend followed by an upward trend, with the lowest points mainly occurring in 2017 and 2018. Notably, Guangdong, as a coastal economically developed province, experienced a decline in agricultural production poverty reduction efficiency below the national average in 2020. This decline is presumed to be primarily due to the severe impact of the epidemic and hindrances to its outward-oriented agricultural products economy. Conversely, Chongqing and Shandong witnessed significant improvements in agricultural production poverty reduction efficiency in 2019–2020.

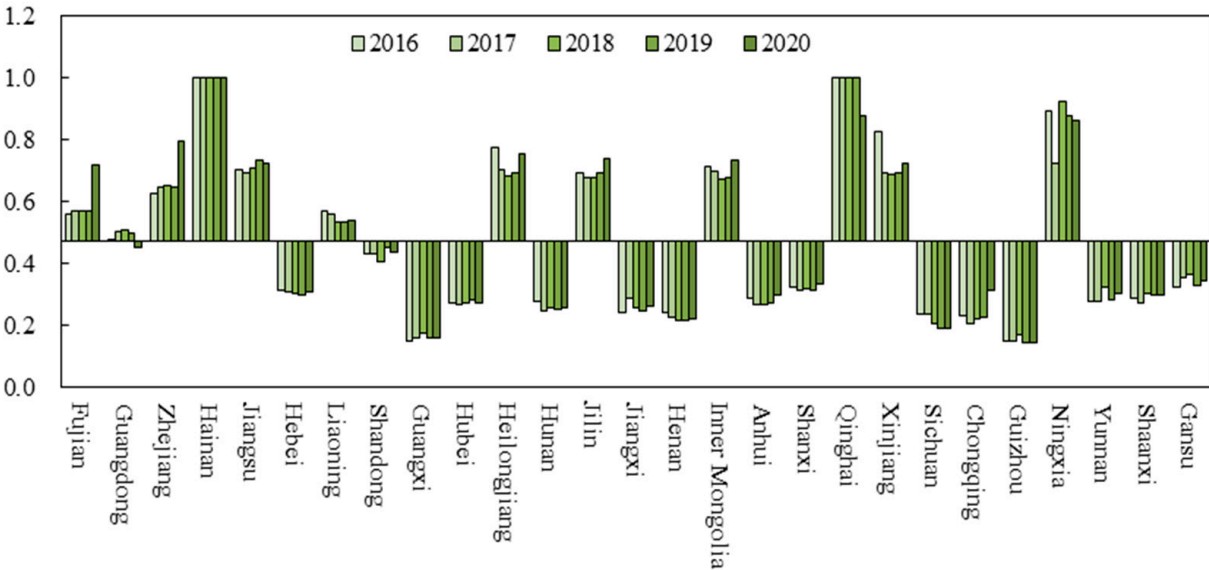

**Figure 4.** Changes in efficiency of agricultural production for poverty reduction by provinces from 2016 to 2020.

On the contrary, the poverty reduction efficiency of agricultural production in Henan and Sichuan experienced a decline for five consecutive years. As China's traditional agricultural and populous provinces, Henan and Sichuan face challenges such as a lower per capita area of arable land compared to the national average, leading to increased initial costs of agricultural inputs and decreased agricultural income. Moreover, the rural labor force is experiencing a massive exodus despite government policies aimed at supporting agriculture, resulting in structural shortages in the rural labor force. These factors have further exacerbated issues related to low-quality and aging agriculture.

As illustrated in Figure 5, the heterogeneity in agricultural production efficiency for poverty reduction across different regions reflects varying changes in combined efficiency values for the three major regions. Over the 2016–2020 period, the average comprehensive efficiency for the eastern, central, and western regions stands at 0.546, 0.415, and 0.455, respectively, with consistent trends in the time series. The eastern region consistently maintains the highest level of comprehensive efficiency, followed by the western and central regions, indicating a higher level of poverty reduction governance in agricultural production in the east compared to the central and western regions, with significant gradient differences.

We utilized ArcGIS 10.8 software to visualize the spatial and temporal evolution of China's agricultural production stage efficiency and poverty reduction stage efficiency in 2016, 2018, and 2020, respectively (Figure 6). In terms of the stage of agricultural production, the comprehensive efficiency of the agricultural production system showed a trend of stepwise decline from north to south. In terms of temporal evolution, the comparative advantage of agricultural production efficiency in the north remained unchanged during the sample observation period, and the agricultural production efficiency in the eastern coastal provinces also showed a substantial increase with the increase in the level of agricultural mechanization, while the agricultural production efficiency in the central region declined.

It is worth noting that agricultural productivity in Zhejiang increased substantially, and agricultural productivity in Gansu improved, while productivity in the traditional agricultural provinces of Hunan and Hebei declined more severely. In terms of the stage of agricultural poverty reduction, the spatial differentiation of agricultural poverty reduction efficiency between regions is obvious. Provinces with higher efficiency in agricultural poverty reduction are mainly concentrated in the eastern coastal region, such as Hainan, Fujian, Heilongjiang, Jiangsu, Guangdong, and Zhejiang. There is more room for improvement in poverty reduction efficiency in the northern regions, with high values occurring in Ningxia and Qinghai. Over time, the spatial variation in poverty reduction efficiency between regions has not widened, with Hebei experiencing a substantial improvement in poverty reduction efficiency and Guangdong experiencing a certain decline in agricultural poverty reduction efficiency as a result of the new Crown Pneumonia epidemic.

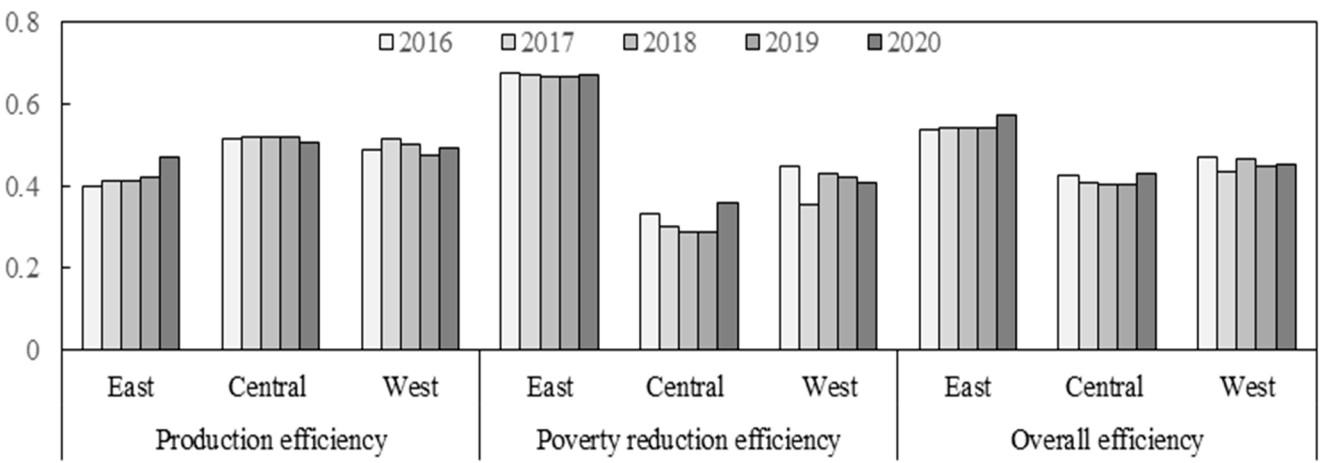

**Figure 5.** Time variation in the efficiency of poverty reduction in regional agricultural production.

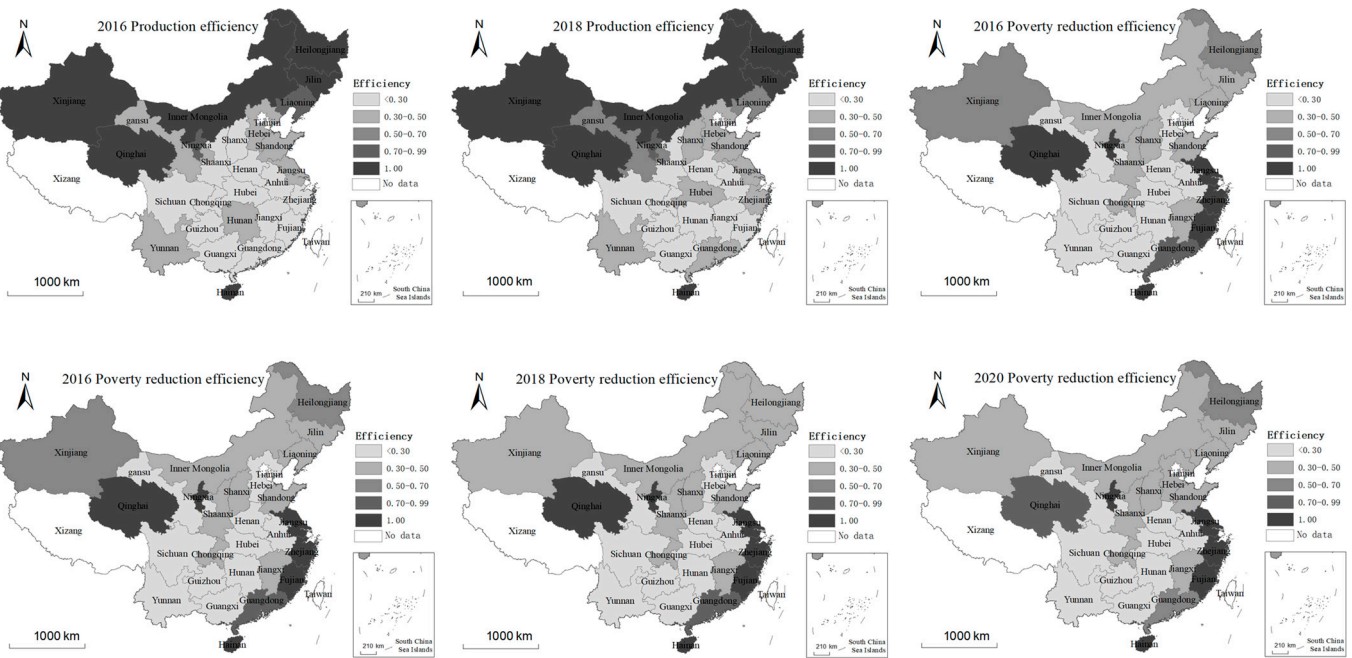

**Figure 6.** Spatial and temporal evolution of efficiency of agricultural production and poverty reduction in China.

### 4.2.4. Efficiency Cluster Analysis for Poverty Reduction in Agricultural Production

To facilitate the comparison of empirical results between the two phases of agricultural production and agricultural poverty reduction, this paper utilized the mean value of the first-phase efficiency (0.480) and the mean value of the second-phase efficiency (0.466) as prime coordinates. Agricultural production efficiency is represented on the horizontal axis, while agricultural poverty reduction efficiency is depicted on the vertical axis. The poverty reduction efficiency level of Chinese agricultural production is categorized into four types: "high-high" type, "low-low" type, "high-low" type, and "low-high" type, as illustrated in Figure 7.

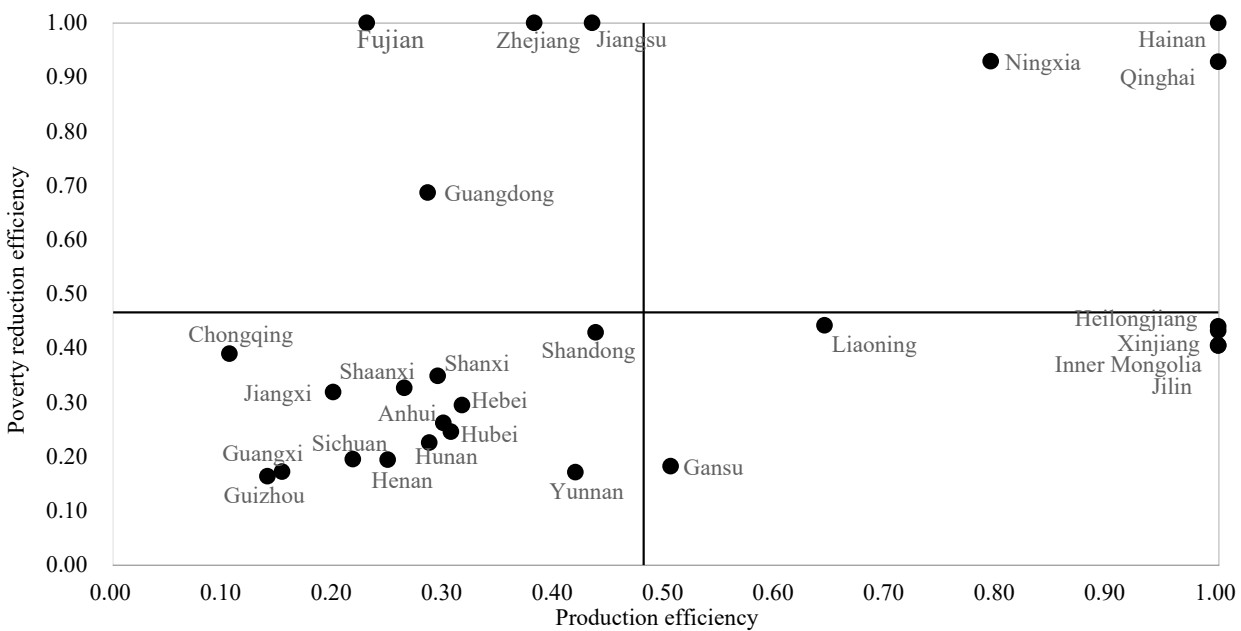

**Figure 7.** Production–poverty reduction efficiency mean value matrix, 2016–2020.

The clustering results indicate that poverty alleviation through agricultural production in China predominantly falls into the "Low-Low" category. Among them, the "High-High" type of agricultural production poverty alleviation, comprising Ningxia, Hainan, and Qinghai in the upper right quadrant, demonstrated relatively high efficiency. However, except for Hainan, which operates under DEA efficient conditions in both agricultural production and poverty alleviation phases, Hainan and Qinghai exhibited lower efficiency in the government poverty reduction phase. This underscores the need for relevant sectors to concentrate on enhancing the efficiency of the second stage in the future while maintaining the efficiency of the first stage. The "Low-Low" category encompasses 14 provinces in the lower left quadrant, representing 51.8% of the total.

### 4.2.5. Input–Output Improvement Analysis of China's Provincial Poverty Reduction Stage

Compared with the traditional DEA, which can only rely on the input perspective or output perspective to improve relative efficiency, the SBM model can analyze the problem from both input and output perspectives (non-directed) and provide input–output indicator optimization analysis for relevant government departments. As shown in Table 4, the poverty reduction efficiency decreased significantly in 2017, so this section takes 2017 as an example (the original table is extensive and is not reflected in the text for reasons of space) and lists the slack values of inputs and outputs (including non-desired outputs) in the poverty reduction stage of 27 provinces.

**Table 4.** Input–output slack variables of the poverty reduction stage of 27 Chinese provinces in 2017.

| Province | Local Financial Expenditure | Central Government Poverty Reduction Fund | Basic Condition Level | Rural Residents Receiving Minimum Living Guarantee |
|---|---|---|---|---|
| Anhui | −2225.13 | −15.30 | 0.00 | −123.32 |
| Fujian | 0.00 | 0.00 | 0.00 | 0.00 |
| Gansu | −1874.80 | −44.26 | 0.00 | −282.20 |
| Guangdong | −3860.71 | −0.02 | 0.00 | −43.85 |
| Guangxi | −3435.14 | −21.95 | 0.52 | −238.01 |
| Guizhou | −3142.14 | −43.09 | 0.00 | −243.37 |
| Hainan | 0.00 | 0.00 | 0.00 | 0.00 |
| Hebei | −2218.36 | −17.17 | 0.00 | −124.21 |
| Henan | −4246.32 | −21.26 | 0.00 | −256.14 |
| Heilongjiang | −38.62 | −15.21 | 0.00 | −65.93 |
| Hubei | −2588.11 | −17.56 | 0.00 | −103.82 |
| Hunan | −3542.05 | −23.78 | 0.00 | −97.26 |
| Jilin | 0.00 | −7.42 | 14.87 | −39.80 |
| Jiangsu | 0.00 | 0.00 | 0.00 | 0.00 |
| Jiangxi | −718.04 | −17.24 | 0.00 | −139.52 |
| Liaoning | −663.69 | −2.99 | 0.36 | −40.39 |
| Inner Mongolia | −53.41 | −15.04 | 0.00 | −82.78 |
| Ningxia | −110.26 | −7.25 | 0.00 | −22.25 |
| Qinghai | 0.00 | 0.00 | 0.00 | 0.00 |
| Shandong | −1910.00 | −1.49 | 0.00 | −117.98 |
| Shanxi | −2097.06 | −10.27 | 0.00 | −91.32 |
| Shaanxi | −2751.46 | −17.76 | 0.00 | −61.83 |
| Sichuan | −4035.29 | −33.51 | 0.00 | −328.83 |
| Xinjiang | 0.00 | −31.43 | 0.00 | −158.60 |
| Yunnan | −3798.18 | −43.10 | 0.00 | −308.48 |
| Zhejiang | 0.00 | 0.00 | 0.00 | 0.00 |
| Chongqing | −1424.72 | −12.50 | 0.00 | −33.45 |
| Mean | −1656.80 | −15.54 | 0.58 | −111.24 |

As a whole, there is little room for improvement in the level of sustainable agricultural development in the provinces, but full technical efficiency can be achieved with an average reduction of USD 165.68 billion in local fiscal expenditures in the provinces and an average reduction of USD 1.554 billion in central government poverty reduction funds in the provinces. In the eastern region, Fujian, Zhejiang, Hainan, and Jiangsu have a poverty reduction efficiency value of 1, so the slack variable is 0, indicating that government inputs in the poverty reduction phase are utilized efficiently. It is worth mentioning that, compared with the eastern region, Qinghai Province has poor natural conditions, closed transportation, a low level of industrial development, a low level of public services, insufficient infrastructure, lagging development of social undertakings, and is not optimal in terms of governmental inputs and policy preferences; however, its poverty alleviation stage efficiency for the period of 2016–2020 is 1, which indicates that the province focuses on improving the efficiency of resource allocation rather than blindly expanding it when considering the policy orientation of agricultural poverty alleviation. On the contrary, Gansu Province, which is also located in the west, has a poverty reduction efficiency of only 0.197, with a redundancy of CNY 187.48 billion and CNY 4.426 billion in local fiscal expenditures and central fiscal poverty alleviation subsidies. The investment of sky-high financial resources not only did not lead to the improvement of poverty reduction efficiency but also caused a serious waste of resources, and financial resources should be allocated rationally. Similarly, in Henan, Guangxi, Sichuan, and Yunnan, it is also more serious. Although these provinces have all successfully realized that all the poor counties in the province have been lifted out of poverty by the end of 2020, attention should still be

paid to preventing the risk of returning to poverty and the phenomenon of returning to poverty. It is important to emphasize that while Jilin is efficient in its use of local fiscal expenditures, it is less efficient in its use of central government funds for poverty reduction. Therefore, in planning for poverty reduction, the government should not only look at the issue from the perspective of inputs but also from the perspective of outputs, focusing on the implementation of resources after they have been invested rather than blindly investing and re-investing them, which would result in an excessive waste of limited resources.

## 5. Conclusions and Policy Recommendations

Since the central special fund for poverty alleviation was renamed as the central financial articulation to promote rural revitalization subsidy in 2021, it changed the scope of use of the special fund. Therefore, the data in this paper, as of 2020, do not take into account the economic results of the COVID-19 pandemic, but this fact will greatly influence future research related to agricultural production–poverty reduction.

In this paper, we utilized a two-stage dynamic network SBM model based on non-expected outputs and simultaneously incorporated central government poverty alleviation funds and local government financial expenditures into the provincial agricultural poverty reduction efficiency evaluation system and decomposed them into production efficiency and poverty reduction efficiency in order to study the poverty reduction efficiency of agricultural production in 27 provinces in China. We found that the two-stage dynamic network SBM model is conducive to reducing the influence of non-desired outputs on the evaluation of the single-stage SBM model, with an efficiency improvement of 97.9%, which further confirms that the two-stage dynamic network model opens the "black box" of the traditional DEA, and is better able to show the whole process of poverty reduction in agricultural production. Additionally, we concluded the following regarding China's poverty reduction efficiency in agricultural production:

Firstly, China's poverty reduction efficiency in agricultural production is average, and it declined substantially in 2017. China's agricultural production poverty reduction efficiency is obviously polarized, and most provinces' agricultural poverty reduction aggregation type is the "double-low" type, indicating that China's agricultural production efficiency and poverty reduction efficiency are not coordinated, and the pulling effect of agricultural production on agricultural poverty reduction is not obvious. In the western and northern provinces, agricultural production efficiency is higher, but the efficiency of government intervention in poverty reduction is lower.

Secondly, in terms of regional differences and spatial and temporal evolution characteristics, there is a more serious East–Central–West gradient difference in the efficiency of agricultural production for poverty reduction. Overall, the eastern region has higher overall efficiency; in terms of stages, production efficiency is higher in the north and lower in the south, in the form of a ladder. The average efficiency is highest in the central region, slightly lower in the western region, and the eastern region has achieved positive growth for five consecutive years, although there is a large gap between the eastern region and the central and western regions; there are both east–west and north–south differences in poverty reduction efficiency, and there is a clear spatial differentiation in poverty reduction efficiency between regions. The high value of poverty reduction efficiency is mainly concentrated in the eastern coastal region, while the low value is concentrated in the central and western regions. The central region's advantages in food production have not really been transformed into advantages in provincial economic development, which may hit the production enthusiasm of the main food-producing areas and even jeopardize national food security.

Thirdly, this paper analyzed the input–output improvement of the poverty reduction stage in 2017 based on slack variables and found that the local financial expenditures in each province can be reduced by an average of CNY 165.68 billion, and the input of the central financial funds for poverty alleviation in each province can be reduced by an average of CNY 1.554 billion, which enables the DEA to be fully effective.

Based on the findings of this paper, several policy implications emerge:

(1) The Chinese Government should adopt a dual-focused approach, emphasizing both agricultural production and rural poverty reduction. This entails enhancing coordination between the efficiency of agricultural production and poverty reduction in provincial areas, aiming to improve the contribution of agricultural development to poverty reduction. Priority should be given to initiatives that promote the development of the western region, accelerate reforms to revitalize old industrial bases in northeastern China, leverage advantages to boost the central region, and establish an effective mechanism for coordinated regional development. Additionally, policies supporting agricultural and rural development should be enhanced across the eastern, central, and western regions. Establishing an information-sharing platform among these regions can help address deficiencies and foster strengths, leveraging the government's macro-control mechanism to guide agricultural production towards poverty reduction. Ultimately, a new model of agricultural and rural development characterized by complementary regional advantages and high-quality development should be pursued to ensure the sustainability of agricultural and rural development.

(2) In response to the variations in agricultural poverty reduction efficiency across provinces, it is advisable for the Chinese Government to refine its poverty reduction strategies tailored to the unique circumstances of northern and southern regions. Leveraging modern science and technology to enhance agricultural productivity and establishing a sustainable long-term mechanism to address the root causes of poverty are crucial steps. As part of advancing sustainable agricultural development within the framework of future rural revitalization initiatives, enhancing the efficiency of agricultural production and the government's poverty alleviation endeavors is paramount to forestall the resurgence of poverty.

(3) The government should promote cross-sectoral integration and synergistic development by fostering collaboration between the agricultural production sector and the poverty reduction sector. There is a need to vigorously promote the adoption of smart and digital agriculture, leveraging technologies such as agricultural big data and blockchain in production and sales processes to reduce costs and enhance efficiency. Additionally, efforts should focus on establishing a modern agricultural science and technology innovation system to increase the scientific content of agricultural practices, enhance mechanization, and improve land and labor productivity.

**Author Contributions:** Conceptualization, J.W.; data curation, J.W. and J.T.; formal analysis, J.W. and J.T.; investigation, J.W.; methodology, Z.F. and J.W.; visualization, J.T.; supervision, Z.F. and J.W.; project administration, J.W.; writing of original draft, J.W.; writing—review and editing, Z.F. and J.W. All authors have read and agreed to the published version of the manuscript.

**Funding:** This research was funded by the Fuzhou Key Research Base of Social Sciences Min Merchants Research Center (2023FZB70) for financial support.

**Institutional Review Board Statement:** Not applicable.

**Informed Consent Statement:** Not applicable.

**Data Availability Statement:** Data are contained within the article.

**Conflicts of Interest:** The authors declare no conflicts of interest. The funding sponsors had no role in the design of the study, the collection, analyses, or interpretation of data, the writing of the manuscript, or in the decision to publish the results.

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
