# Peer review of "Assessing the Drivers of Sustained Agricultural Economic Development in China: Agricultural Productivity and Poverty Reduction Efficiency"

_sustainability, doi:10.3390/su16052073_

Round 1

Reviewer 1 Report

Comments and Suggestions for Authors

The proposed research method is of considerable interest in relation to the peculiarities of the organization of state socio-economic policy in China. But there are some omissions, that decrease the quality of reasoning and validity of conclusion.

1. The statistical data in Introduction (pp. 1-2) has no references. So, we coudn't evaluate precondition of the basic thesis of the researce.

2. In Literature review authors didn't represent main theoretical concepts analyzed in the article: poverty and sustainable agricultural development. In total significant part of the rewiew consist discription of this researce's results, but not the results of other authors.

3. In Research methodology the first stage of the model hasn't describe. Also there aren't clearly representation of the data's sources. 

4. The recommendations submitted in Conclusion don't take into account the  economical results of pandemic COVID-19. This can be explained by the fact that only data up to 2020 was used for the study. But this fact significantly influence to the prognostic function of the research.

Author Response

Please see the attachment, Thank you very much.

Reviewer 2 Report

Comments and Suggestions for Authors

Thhis is a review of a manuscript titled: “Assessing the drivers of sustained agricultural economic development in China agricultural productivity and poverty reduction efficiency”.

Dear authors. I hope this find you well. This is a very interesting article that presents information about public policies that in the last years have helped reduced poverty in Chines provinces

The title needs to be clearer. Usually what other authors in similar articles write would be something like:  Assessing the drivers of sustained agricultural economic development in China: agricultural productivity and poverty reduction efficiency.

Line 15. Mention the qualitative and quantitative measurements.

Line 23. Please include the northern and southern provinces.

Line 42. Include what CPC stands for.

Line 48. What do you refer to in Chinese characteristics.

Line 52. Please include the differences between the levels of poverty.

Line 65. How do the central government’s financial funds for poverty alleviation work?

Line 92. DBSM and DEA include in parenthesis what stands for.

Line 124. How traditional agricultural provinces work?

Line 154. Have you found existing literature about assessment in other places that could help to understand your research?

Line 284. What type of crops

Please check about table 1 and table 2, in the document have the same number.

Table 1. Please include in expected output the conversion to US dollars to reach a broader audience.

Line 341. It would be convenient to make a table about the characteristics and differences between provinces (e.g. km2, number of habitants). You could include information that you present on line 363.

Line 411. What do you mean by production efficiency?

Figure 7. The maps can be benefited by increasing the size in a 2 by 6 organization and including colours.

Line 492. Could you elaborate on Qinghai characteristics?

Line 553. Do you have a good example of this policy implication?

Best regards,

Author Response

(The authors gave the same response as above.)

Reviewer 3 Report

Comments and Suggestions for Authors

The manuscript “Assessing the drivers of sustained agricultural economic development in China agricultural productivity and poverty reduction efficiency” submitted by Fang and coworkers analyzed the performance of China’s agricultural production poverty reduction efficiency, particularly the differences between different regions. Methods such as the DSBM model and the network DEA model has been employed to analyze the agricultural production efficiency, the poverty reduction efficiency and the total efficiency of agricultural production and poverty reduction. Overall, the study was carefully designed and the data and discussions support their main conclusion. The logic of the manuscript is clear and good. The originality is not very high, but it could provide valuable insights for scholars and policy makers in the related fields. Considering these merits, I’d like to recommend the acceptance of this manuscript for publication on Sustainability. Two advices to improve the manuscript: 1) The authors should briefly introduce the analytical methods they used in this study. 2) the full name of “DSBM model”, “DEA” model” should be provided in the first appearance.

Author Response

(The authors gave the same response as above.)

Round 2

Reviewer 1 Report

Comments and Suggestions for Authors

The changes made greatly improved the quality of the article and the understanding of the research methodology as well as the authors' logic.

Reviewer 2 Report

Comments and Suggestions for Authors

Dear authors,

Thank you for the effort of making the corrections of the manuscript. I suggest accepting the article in present form.  

Best regards,